

# A novel permanent gauge-cam station for surface flow observations on the Tiber river

Flavia Tauro[1], Andrea Petroselli[2], Maurizio Porfiri[3], Lorenzo Giandomenico[4], Guido Bernardi[4], Francesco Mele[5], Domenico Spina[5], and Salvatore Grimaldi[1,3]

[1]Dipartimento per l'Innovazione nei Sistemi Biologici, Agroalimentari e Forestali, University of Tuscia, Viterbo 01100, Italy.
[2]Dipartimento di Scienze Agrarie e Forestali, University of Tuscia, Viterbo 01100, Italy.
[3]Department of Mechanical and Aerospace Engineering, New York University Tandon School of Engineering, 11201 Brooklyn, NY, USA
[4]CAE S.p.a., San Lazzaro di Savena 40068, Italy
[5]Centro Funzionale Regione Lazio, Roma 00145, Italy

*Correspondence to:* Salvatore Grimaldi
(salvatore.grimaldi@unitus.it)

**Abstract.** Flow monitoring of riverine environments is crucial for hydrology and hydraulic engineering practice. Besides few experimental implementations, flow gauging relies on local water level and surface flow velocity measurements through ultrasonic meters and radars. In this paper, we describe a novel permanent gauge-cam station for large scale and continuous observation of surface flows, based on remote acquisition and calibration of video data. Located on the Tiber river, in the center of Rome, Italy, the station captures one-minute videos every 10 minutes over an area oriented along the river cross-section of up to $20.6 \times 15.5\,\mathrm{m}^2$. In a feasibility study, we demonstrate that accurate surface flow velocity estimations can be obtained by analyzing experimental images via particle tracking velocimetry (PTV). In medium illumination conditions ($70 - 75\,\mathrm{lux}$), PTV leads to velocity estimations in close agreement with radar records and is less affected by uneven lighting than large scale particle image velocimetry. Future efforts will be devoted to the development of a comprehensive testbed infrastructure for investigating the potential of multiple optics-based approaches for surface hydrology.

## 1 Introduction

Understanding the kinematic organization of natural water bodies is central to hydrology and environmental engineering practice (LeCoz et al., 2010). Reliable flow velocity estimations are essential to comprehend flood generation and propagation mechanisms. Continuous flow observations are often required in the investigation of erosion dynamics, sediment transport, and drainage network evolution (Hrachowitz et al., 2013; Montanari et al., 2013). In engineering practice, flood warning systems largely rely on real-time discharge measurements, and flow velocity monitoring is important for the design and management of hydraulic structures, such as reservoirs and hydropower plants (Kantoush et al., 2011).

Traditionally, gauging stations have been equipped with water level meters, and stage-discharge relationships (rating curves) have been established through few direct discharge measurements (Creutin et al., 2003). Only in rare instances, monitoring stations have integrated radar technology for local measurement of surface flow velocity (Costa et al., 2006; Fulton and Os-





trowski, 2008). Establishing accurate rating curves depends on the availability of a comprehensive range of discharge values, including measurements recorded during extreme events. However, discharge values during high-flow events are often difficult or even impossible to obtain, thereby hampering the reliability of discharge predictions.

In the past ten years, promising advancements in flow monitoring have been put forward due to fully remote optics-based velocimetry techniques (Muste et al., 2008; Tang et al., 2008; Tauro et al., 2012a, b). Such methodologies enable the estimation of the surface flow velocity field over extended regions from the relative motion of naturally occurring debris or floaters dragged by the current (Fujita et al., 1997). Surface flow velocity measurements are typically related to depth-averaged velocity, and discharge is computed from information on the cross-section (Alessandrini et al., 2013; Chiu, 1991; Farina et al., 2014; Jodeau et al., 2008; Moramarco et al., 2004; Tazioli, 2011). Among optical approaches, large scale particle image velocimetry (LSPIV) is an extension of classical particle image velocimetry (PIV) (Adrian, 1991; Raffel et al., 2007). It is based on the acquisition of pairs of images by a digital camera at known acquisition frequency. Such image pairs are then orthorectified, georeferenced, and a high-speed cross-correlation algorithm is implemented to generate velocity maps. The accuracy of the method is highly dependent on the occurrence and visibility of floaters as well as on illumination conditions (Hauet et al., 2008a).

LSPIV implementations have allowed the characterization of flow patterns in river estuaries (Bechle et al., 2012), lakes (Admiraal et al., 2004), and large scale riverine environments (Hauet et al., 2009). Most frequently, measurements are conducted through ad hoc installations, which include an angled camera to capture extended fields of view and a processing unit to estimate the velocity field (Bradley et al., 2002). Several observations have been performed using mobile configurations, where frame acquisition is enabled by portable stand-alone instruments or trucks (Kim et al., 2008; Dermisis and Papanicolaou, 2009). In (Hauet et al., 2008b), continuous and real-time river monitoring has been demonstrated through a fixed LSPIV installation located on a roof nearby the Iowa River. To the best of our knowledge, this implementation is the only fixed LSPIV measurement station for continuous monitoring of a river flow. Similar implementations placed underneath bridges and boardwalks have enabled the observation and analysis of selected flood events (Muste et al., 2011; Tsubaki et al., 2011).

These studies have fostered further efforts toward the refinement of optics-based methods for accurate surface flow measurements. In (Tauro et al., 2014b), a modified setup has been proposed to allow for remote digital image acquisition and calibration based on the use of low-power laser modules. This implementation has proved convenient for observations in difficult-to-access environments and during flood events; however image processing through LSPIV has been found to be highly affected by the occurrence and spatial distribution of tracers. Motivated by promising experimental findings obtained with the mobile setup, herein, we propose a permanent optics-based sensing platform for surface flow observations. Specifically, we present an innovative experimental gauge-cam station featuring remote image acquisition and calibration for continuous observation of surface flows in the Tiber River at Ponte del Foro Italico, in the center of Rome, Italy. The gauge-cam station enables the acquisition of massive video data throughout the year, providing a remarkable testbed to assess the feasibility and validate optics-based flow measurement approaches.

Since December 2014, the gauge-cam station has acquired digital videos that display variable hydraulic regimes, ranging from flood events to low waters, illumination conditions, and sediment loads. In this paper, we focus on three videos recorded in January and February 2015, whose analysis is undertaken through particle tracking velocimetry (PTV) and LSPIV to generate



surface flow velocity field maps in the $20.6 \times 15.5\,\mathrm{m}^2$ field of view. Both sets of findings from video data are compared to radar measurements and discussed for future research investigations.

The rest of the paper is organized as follows. In Section 2, the pre-existing local gauging station and the proposed novel measurement installation are described. In Section 3, we present the research objectives underlying the design of the gauge-

cam station along with possible research directions enabled by its installation. In Section 4, a representative experimental video is analyzed with two different optics-based approaches. In Section 5, we highlight advantages and limitations of our novel gauge-cam station and identify future possible ameliorations. Section 6 summarizes concluding remarks.

## 2  Experimental Station

The experimental station is located in the urban tract of the Tiber river at Ponte del Foro Italico in the center of Rome, Italy

(41°56′22.7″ N 12°29′09.2″ E), see Figure 1. An existing monitoring station managed by Centro Funzionale - Regione Lazio is located on the same bridge. It includes a ULM 20 ultrasonic meter by CAE S.p.a., which records water levels proximal to the midspan of the bridge every 15 minutes. Further, an RVM20 speed surface radar sensor by CAE S.p.a. operating in the $0.30$ to $15\,\mathrm{m/s}$ velocity range with an accuracy of $\pm 0.02\,\mathrm{m/s}$ records surface velocity every 15 minutes over an area of few squared centimeters.

The gauge-cam station is based on the advanced Multi-Hazard System (Mhas) technology developed by CAE S.p.a. for integrated environmental monitoring (www.cae.it). It comprises two units: a central control apparatus and a sensing platform, see Figure 2. The control unit is equipped with several input ports to coordinate multiple sensing modules. Specifically, the unit interfaces all measurement devices, runs the sensors, receives, and stores real-time data. A solar-cell rechargeable battery technology guarantees the gauge-cam station operation for considerable periods of time and in case of interrupted power. Data

are stored in a datalogging terminal that allows for locally visualizing measurements and provides rapid computing capacity thanks to a dual-core processor and the embedded Linux operating system. Connection to the unit is established from any web-based device through http-communication. By accessing the control unit, users control the station actions, set the events' chaining and time table, edit recording settings, and overlook and potentially download current images. Video storage is enabled through a 1 Tb solid state external hard drive, which features ext4 journaling file system for high storage limits.

The sensing unit is based on the portable prototype for LSPIV observations developed in (Tauro et al., 2014b). Specifically, it is connected to the control unit and suspended underneath the bridge through an aluminum bar at an elevation of approximately $15\,\mathrm{m}$ from the water surface, see Figure 2c. All sensors are enclosed in aluminum cases and connected to a $1\,\mathrm{m}$ horizontal aluminum bar. At the center of the horizontal bar, a digital camera is hosted, whereas two $< 20\,\mathrm{mW}$ green lasers ($532\,\mathrm{nm}$ in wavelength) are installed at the two ends of the bar, $50\,\mathrm{cm}$ apart from the camera axes. The lasers are diode-pumped solid-state

continue-wave modules and are operated at $3\,\mathrm{V}$ and $< 300\,\mathrm{mA}$ (http://www.apinex.com, 2015). Each module is encased in a $35\,\mathrm{mm}$ in length and $12\,\mathrm{mm}$ in diameter brass cylinder.

The digital recording system is a Mobotix FlexMount S15 weatherproof internet protocol camera (www.mobotix.com). It is inherently designed for outdoor acquisitions and long-time operation. Specifically, it comprises two miniature optical sensor



modules, that are connected to the camera housing through a sensor cable. Two separate rooms host the optical sensors and lenses. The L25 ($82°$ angle of view and $4\,mm$ focal length) and L76 ($27°$ angle of view and $12\,mm$ focal length) optical sensors are located with their axes perpendicular to the water surface to capture the central portion of the river. The higher angle of view (L25) sensor allows for acquiring a larger area of the river surface, whereas the lower angle of view (L76) sensor synchronously

captures finer details in the center of the L25 sensor field of view.

Temporally-resolved surface flow observations at high temporal resolutions are obtained by setting the digital recording system to capture one-minute long videos every 10 minutes. The frame acquisition frequency during the recordings is automatically set based on the illumination conditions sensed by the optical sensors, and is limited to a maximum of $12\,Hz$. Image resolution for both optical sensors is set to $1024 \times 768\,pixels$. To enable remote image calibration, the laser modules are oper-

ated for $20\,s$ at the beginning of each video recording. Videos are stored in the MxPEG audio/video container format, which guarantees the synchronous stream of good quality images at efficient compression. However, the encoder requires dedicated proprietary software for video conversion to avi and image extraction. Videos are stored through a nestled folding system in the external hard drive.

Specifically, in each folder, several types of data are stored using the MxPEG encoder. Video data are stored as jpg files (not

exceeding $17\,Mb$ each), the first frame of each video is saved as a lighter jpf file, and information on the recording settings are contained in light INFO.jpg files (readable with text editing software). In Figure 3, a representative video frame is displayed. In the top left, the station identification name (Foro Italico) and illuminance intensity (in $lux$) for both optical sensors are reported. In the top right, date and time of the day (in the format: year - month - day, time zone abbreviation, hours:minutes:seconds) are illustrated. In the bottom left and right, acronyms related to the station internal protocol for sensor triggering and recording are

shown.

## 3    Research Objectives

The overarching objective of installing a permanent gauge-cam station is to demonstrate a novel, transformative approach for noninvasive estimation of flow discharge in riverine environments. In this vein, the existing gauging station at ponte del Foro Italico (which is regularly monitored by Centro Funzionale - Regione Lazio) has been empowered with optical sensors for a

thorough validation and comparison of surface flow velocity measurements. Although numerous technical contributions point out the advantage of using image-based technology against traditional instrumentation (LeBoursicaud et al., 2015; Muste et al., 2011; Tauro, 2015), optical methodologies are scarcely used in engineering practice.

While several studies investigate the feasibility of using mobile (Dramais et al., 2011) and aerial optical platforms (Fujita and Hino, 2003; Fujita and Kunita, 2011; Tauro et al., 2015b) for enhanced versatility, stationary implementations should offer

more consistent experimental settings. This results in simplified image preparation; for instance, image matching is rarely required before LSPIV processing of videos captured from stationary implementations (Tauro et al., 2014b). In the future, we plan on testing video data recorded at the gauge-cam station through an array of image-based algorithms, spanning from LSPIV to particle tracking velocimetry (Tang et al., 2008), long-term tracking (Pervez and Solomon, 1994), and optical flow (Quénot




et al., 1998). In addition, classification of such a large data base will leverage the application of unsupervised machine learning procedures (Tauro et al., 2014a).

We aim at investigating the feasibility of using fully-autonomous optical methods for the kinematic characterization of surface flows over extended water bodies. As anticipated in (Tauro et al., 2014b) and further supported in this work, image-

based algorithms such as LSPIV tend to be highly affected by varying flow settings. To overcome this issue, we plan on thoroughly studying the relationship between time variations of the velocity field and flow conditions. We will explore the effect of the following parameters on surface flow estimation: illumination conditions (this will be enabled by the camera built-in real-time light sensors), flow and meteorological conditions (hydrometeorological conditions are provided from the local gauging station), sediment loading, and floaters' visibility and space distribution.

Continuous and remote video acquisitions will also provide a comprehensive testbed for assessing the efficacy of optical methods in surface hydrology. Indeed, the availability of such a wide data base will enable the quantification of uncertainty of image-based surface flow measurements. We expect the gauge-cam station to aid in the definition of optimal operational settings for different optical algorithms. Optimal experimental conditions may be condensed in lumped indices to inform the use of optical methods in environmental settings.

## 15  4   Case Study

The potential of the gauge-cam station to provide spatially distributed surface flow velocity maps is demonstrated through the analysis of a set of videos recorded on January 2nd, 2015, from 07:50:01 to 07:51:01. During the one-minute registration, eight video files were saved for a total of more than $119\,\mathrm{Mb}$ of data (MxPEG format). During the selected time, illumination intensity varied from 93.3 to $100.0\,\mathrm{lux}$ for the left-side optical sensor and from 70.8 to $76.7\,\mathrm{lux}$ for the right-side optical sensor.

Naturally occurring circular white floaters (from 10 to $20\,\mathrm{pixels}$ in diagonal as visible in the left-side images) were scattered in the entire field of view. At the time of the registration (measurements relative to 07:45:00 and 08:00:00), the local ULM sensor recorded levels between 1.45 and $1.46\,\mathrm{m}$. Radar measurements were $0.87\,\mathrm{m/s}$ and $0.88\,\mathrm{m/s}$ at 07:45:00 and 08:00:00, respectively.

### 4.1   Image Processing

Surface flow velocity field was reconstructed using two approaches: PTV and particle image velocimetry (PIV). Prior to flow velocimetry, videos were prepared as follows. Video files in the MxPEG format were opened with the MxControlCenter camera propriety software and separate avi files were saved for the left-side and right-side optical sensors, see Figure 4a and c. Such avi files were then decompressed into bmp $2048 \times 768\,\mathrm{pixels}$ images using in-house codes for Matlab environment, and the acquisition frequency was computed for each video. On average, images reported herein were taken at $9.28\,\mathrm{Hz}$ by the Mobotix

camera. Frame borders were then trimmed to obtain $1024 \times 768\,\mathrm{pixels}$ images and the sole green channel was retained for analyses.



To emphasize lighter particles against a dark background, images were gamma-corrected to darken midtones (Forsyth and Ponce, 2011), see Figure 4d. Right-side optical sensor images were fish-eye undistorted using the Adobe Photoshop "Lens correction" filter (automatic geometric distortion correction with distort amount set to 88 and image size set to 82). Finally, both sets of images were processed by mean intensity subtraction to further highlight the presence of floaters against homoge-

5 neous backgrounds, see Figure 4b and d. Image calibration was based on the lasers' trace onto the water surface. Specifically, calibration factors (to convert from pixel to metric velocities) were determined through calibration images, where the distance between lasers and the camera frame acquisition rate were taken as inputs.

Sequences of 561 $1024 \times 768$ pixels bmp images from the left and right-side sensors were analyzed using PTVlab (Brevis et al., 2011). Particle detection was enabled through a Gaussian mask procedure (correlation threshold set to $0.5$, radius set to

10 $9$ pixels, and intensity threshold to $100$). Tracking was based on cross-correlation between pairs of subsequent images, whereby the interrogation area was set to $20$ pixels and $40$ pixels for the right and left-side images, respectively. Further, minimum correlation was set to $0.4$ and similarity among neighbor windows to $20\%$. Velocities along the current and perpendicular to the flow were computed at the nodes of a $10 \times 10$ pixels-cell grid overlayed on images. A total of 560 grids was computed and averaged to obtain surface flow velocity maps.

Particle image velocimetry was executed with the edPIV software (Gui, 2013) on a sequence of 561 images for the left-side sensor and on 187 subsampled ($3.09$ Hz) frames for the right-side sensor. For both sensors, images were resampled to reduce resolution to $640 \times 480$ pixels. Grid and interrogation window sizes were set to $16 \times 16$ pixels and $32 \times 32$ pixels, respectively. Surface flow velocity maps were generated by averaging results in time.

### 4.2 Velocimetry Results

Figure 5 displays LSPIV and PTV time-averaged maps for the left and right-side videos. White areas in PTV maps are due to the absence of floating objects detected and tracked by the algorithm. As expected, the right-side PTV map, Figure 5d, presents a higher density of tracked particles (and, therefore, smaller white areas) due to the larger field of view. Minimum, maximum, and average velocities are reported in Table 1 along with standard deviations. Such values are obtained by analyzing time-averaged maps in Figure 5. On average, LSPIV results are much lower than PTV estimates. In addition, maps in Figures 5a

and c are highly affected by illumination conditions. Indeed, both maps display extremely low velocities in the portion of the field of view that is directly exposed to light and is not covered by the bridge shadow. This effect is only partially mitigated in Figure 5c, where the larger field of view results in slightly higher velocities even in regions lying outside the bridge shadow.

With respect to the left-side map, velocities computed by averaging over the entire field of view are equal to $0.35$ m/s and $0.83$ m/s for the LSPIV and PTV analyses, respectively. Right-side velocities averaged over the entire region of interest are

30 equal to $0.50$ m/s and $0.68$ m/s for the LSPIV and PTV analyses, respectively. Compared to measurements from the RVM20 radar, PTV average velocities are closer to benchmark values than LSPIV results. This finding is in agreement with similar LSPIV implementations (Tauro et al., 2015c) in a smaller scale mountainous stream.



## 5   Discussion

LSPIV processing of the selected recording results in considerably lower velocity estimates than radar measurements and PTV values, as low as nearly one third of radar data. This fact is consistent with previous studies (Tauro et al., 2014b, 2015a), and is mainly attributed to the high sensitivity of LSPIV to illumination conditions and to the irregular presence and distribution

of floaters in the field of view (Tauro et al., 2015a). In our case study, the river mirror-like surface is highly detrimental for velocity estimation. Indeed, recordings from both optical sensors consistently present regions at unrealistically low velocity in the upper half of the images. Further, the irregular size of the floaters and their discontinuous transit lead to considerable underestimations. According to (Tauro et al., 2015c), velocity estimation closer to real values may be obtained by analyzing only image sequences depicting the stationary occurrence of tracers.

On the other hand, PTV findings are promising and in agreement with radar velocities, within a 20% difference. In particular, estimates obtained from the left-side video are close to radar data, whereas the average velocity decreases in the right-side video. This is attributed to the poorer visibility of the floaters in the right-side frames, where a larger region is captured at the initial resolution of $1024 \times 768$ pixels, and images are subject to multiple preprocessing operations, including fish-eye removal, before velocimetry analysis. Based on our findings, PTV could serve as a good alternative to LSPIV in case of spatially

inhomogeneous tracer seeding.

With regards to illumination conditions, both approaches may suffer from excessive or insufficient light as the visibility of tracers is severely impacted. We demonstrate the adverse impact of illumination by presenting two extreme case studies. A first video is recorded on February, 7th, 2015, from 13:21:39 to 13:22:39 at an illumination intensity of 6920 lux. The second video is captured on January 2nd, 2015, from 17:10:02 to 17:11:02 at an average illumination intensity of 2 lux. In both

cases, naturally occurring floaters were transiting in the field of view but their visibility was hampered by light intensity. In Table 2, input parameters for PTV and LSPIV analyses are reported for both sets of experiments. Surface flow velocity maps are generated for both sets of data by averaging results in time. Related statistics are illustrated in Table 3, where radar data are also reported for comparison.

Based on our findings, both image-based algorithms are negatively impacted by adverse illumination conditions. In case

of excessive light, left-side maps result in consistently underestimated velocities. In case of right-side observations, both algorithms lead to unrealistically high average velocities and standard deviations. This behavior is attributed to the abundant water reflections that irregularly appear on the water surface and are erroneously treated as surface tracers by the algorithms. In scarce illumination conditions, both approaches lead to extremely low velocity values. High standard deviation in the right-side LSPIV analysis is attributed to inaccuracy in the correlation procedure due to the homogeneous dark background. High

deviations in PTV analyses are instead related to the disappearance of the tracers during their transit in the field of view. Both tests suggest that optics-based algorithms are highly dependent on tracers visibility and, therefore, on illumination conditions. In this context, information on light intensity are continuously monitored by the station and may guide future analyses.

This preliminary study demonstrates the versatility of the gauge-cam station and its advantages with respect to traditional instrumentation. Specifically, in the first semester of 2015, almost 1 Tb of data have been recorded for a total of more than



400 h of high-quality videos. This extremely rich data base depicts the surface flow of the Tiber river for a multitude of variable illumination conditions (from nightime to almost 7000 lux) and flow regimes (for instance, two flood events occurred in February and April). Throughout the operation of the gauge-cam station, minimal maintenance and operational costs were required for supplying power, hard disk periodic replacement, and internet connectivity. Compared to traditional flow velocity instrumentation, the apparatus enables fully remote measurements over a wide range of velocities: due to the large field of view captured by the right-side optical sensor, velocities up to few meters per second can be reconstructed. Further, the system of lasers enables remote and accurate image georeferencing in case of variable flow conditions.

Conversely, a standard procedure for operating the gauge-cam station and handling video data is yet to be achieved. The mirror-like surface of the Tiber river at this particular location poses several problems to the application of correlation-based algorithms. Further, the rather extended cross-section hinders efficient artificial seeding. Such criticalities will demand advanced image enhancement to highlight the presence of traceable objects. Another considerable limitation toward a fully unsupervised operation of the gauge-cam station is the proprietary video encoder, which requires the use of the dedicated camera software, thereby preventing rapid data analysis. In this respect, several video formats and encoders will be tested to facilitate image extraction. Finally, a user-friendly platform will be developed for the generation of surface flow velocity maps and quick comparison to radar data.

## 6   Conclusions

In this paper, we presented a novel fixed gauge-cam station for flow velocity observations in riverine settings. The station is based on the remote acquisition and calibration of high quality images. Maps of the surface velocity field were generated over extended areas up to $20.6 \times 15.5\,\mathrm{m}^2$ by applying optics-based algorithms on video data. Since December 2014, the station has allowed the recording of more than 1 Tb of video data. In a representative case study, we analyzed one minute of video data through two different optics-based techniques: LSPIV and PTV. As expected from previous studies, LSPIV underestimates surface flow, while PTV measurements are in agreement with radar data.

The proposed installation offers a viable platform for thorough testing and validation of image-based algorithms under a wide array of variable illumination and flow conditions. In future studies, we plan on investigating the optimal operational settings for several vision approaches, including LSPIV, PTV, optical flow, and machine learning. Particular care will be devoted to the analysis of flood events during which river velocity estimations may be particularly challenging.

**Data availability**

Research data are freely available at doi: 10.4121/uuid:68ef90c2-d9da-4511-a7ba-d21f68769e03

*Acknowledgements.*  This work was supported by the Ministero degli Affari Esteri project 2015 Italy-USA PGR00175 and by the UNESCO Chair in Water Resources Management and Culture.



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



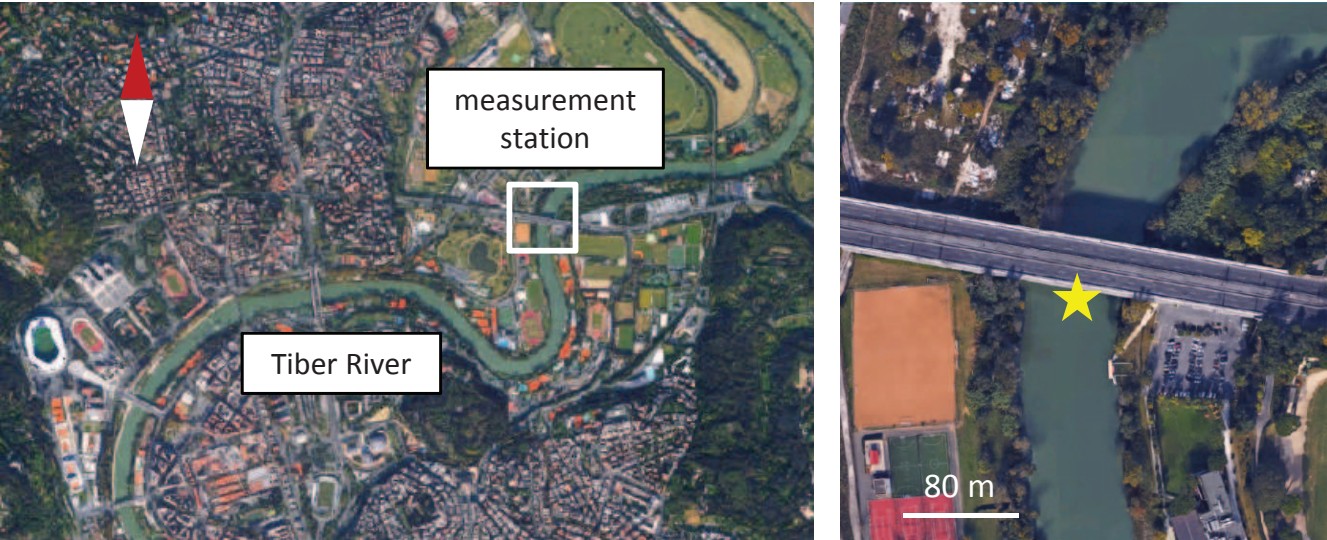

**Figure 1.** Left, view of the urban tract of the Tiber river: the white box encloses Ponte del Foro Italico. Right, close-up view of the measurement site: the yellow star indicates the location of the gauge-cam station.

**Table 1.** Synoptic table for time-averaged velocity maps in Figure 5. Values indicate minimum ($v_{\min}$), maximum ($v_{\max}$), mean ($\overline{v}$) velocities, and standard deviations ($\sigma_{v}$) computed over time-averaged maps, and radar velocity ($v_{R}$).

| | $v_{\min}$ [m/s] | $v_{\max}$ [m/s] | $\overline{v}$ [m/s] | $\sigma_{v}$ [m/s] | $v_{R}$ [m/s] |
|---|---|---|---|---|---|
| 01/02 07:50 | | | | | 0.87 |
| **LEFT** | | | | | |
| PTV | 0 | 2.96 | 0.83 | 0.28 | |
| LSPIV | 0 | 1.40 | 0.35 | 0.40 | |
| **RIGHT** | | | | | |
| PTV | 0 | 2.07 | 0.77 | 0.18 | |
| LSPIV | 0 | 1.94 | 0.57 | 0.58 | |

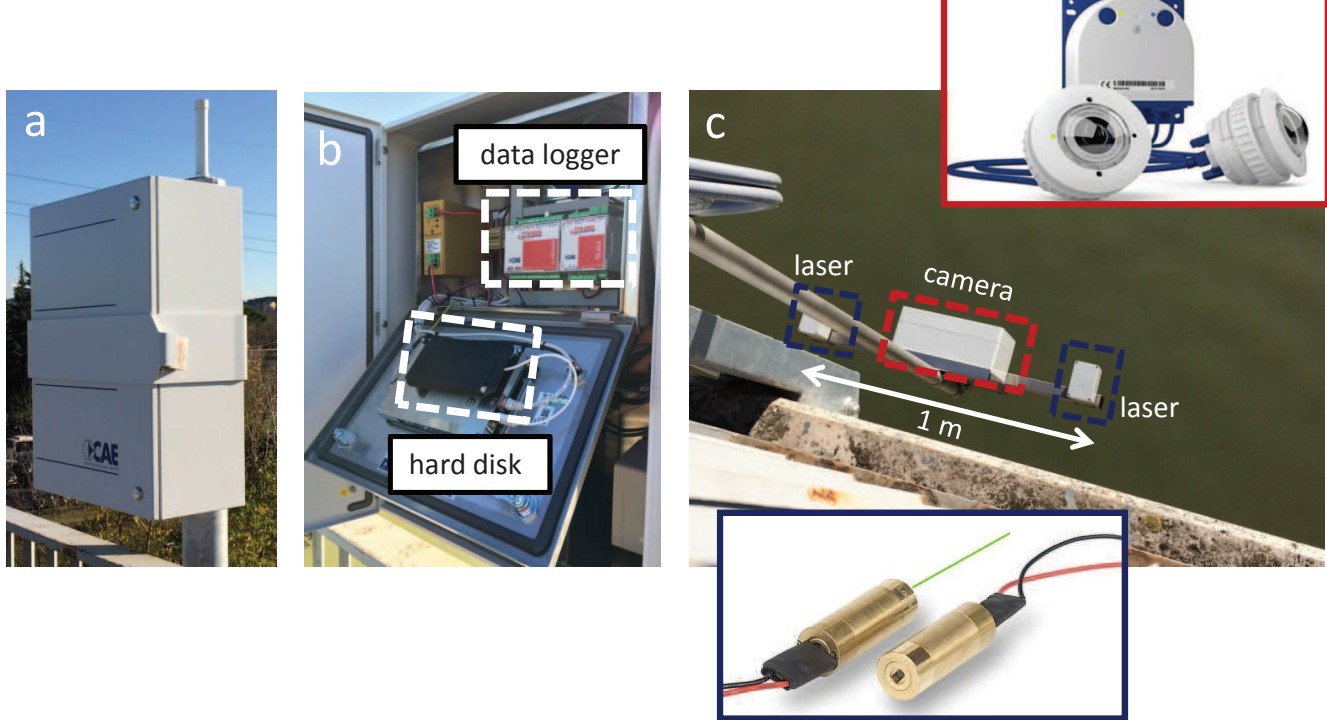

**Figure 2.** Components of the gauge-cam station including: a) the control unit, b) datalogging terminal, and c) sensing unit with camera and lasers.

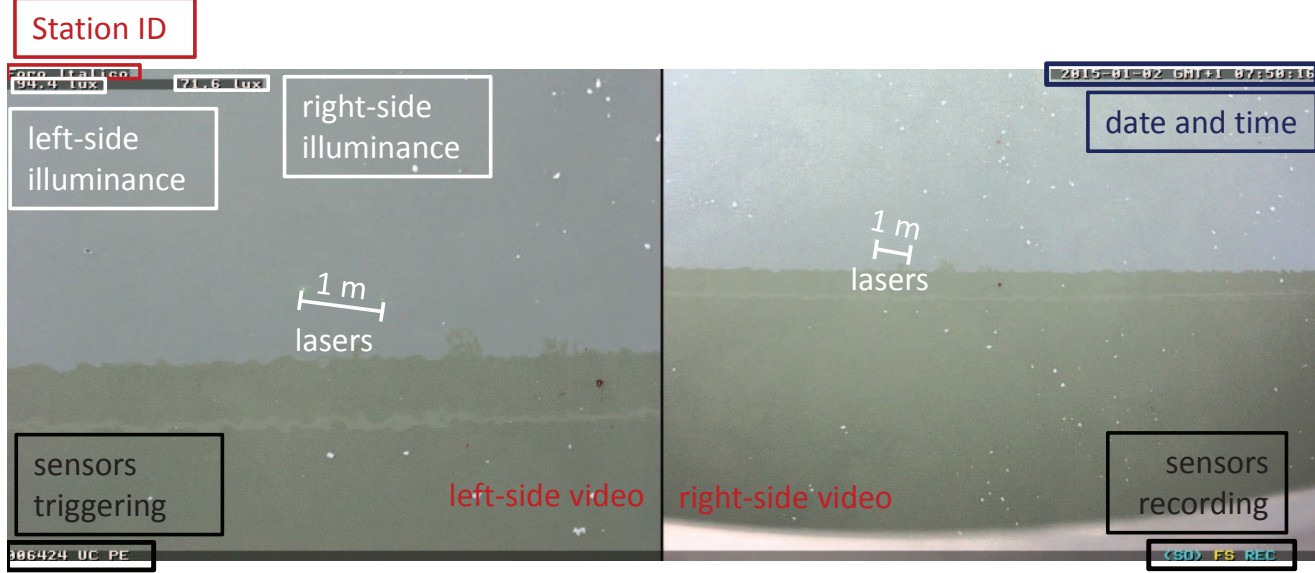

**Figure 3.** Representative experimental video frame captured from the gauge-cam station. Left, field of view recorded with the left-side optical sensor and right, with the right-side optical sensor.

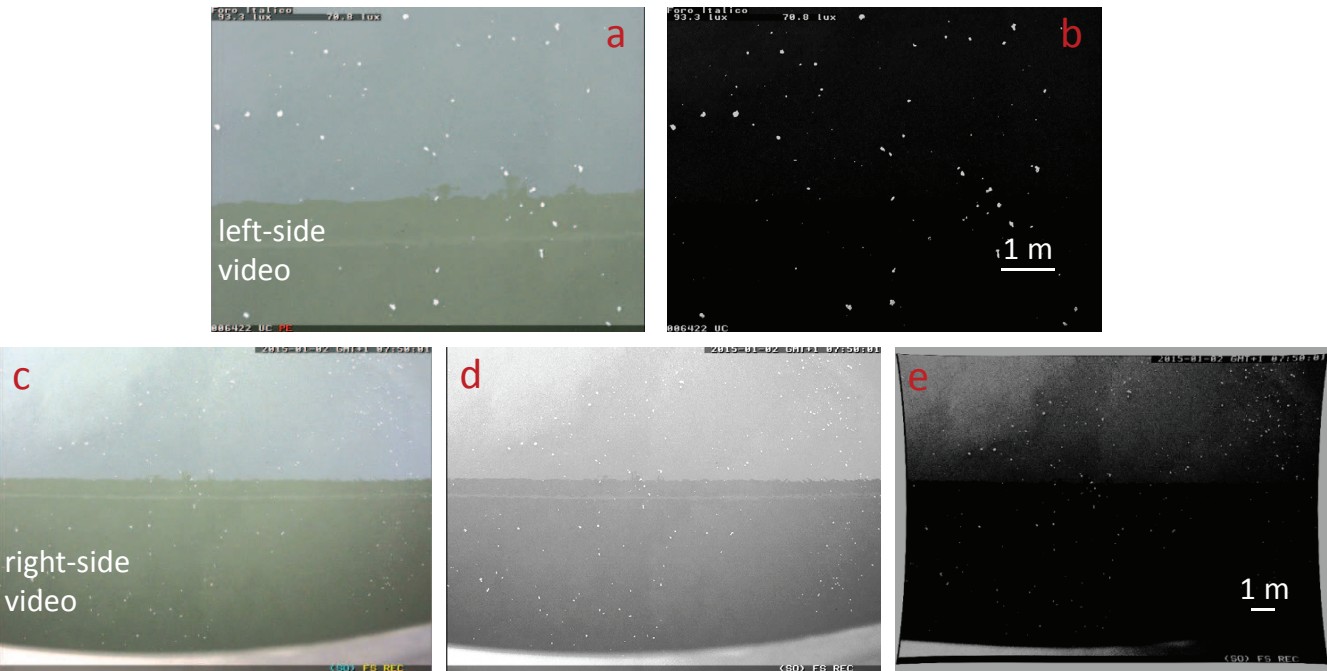

**Figure 4.** Image processing for the left-side (a and b) and right-side (c, d, and e) videos. Left-side videos are: a) isolated and b) filtered and midtone-corrected. Right-side videos are: c) isolated, d) filtered, and e) fish-eye undistorted and midtone-corrected.





**Figure 5.** Experimental findings for the analyzed case study: a) LSPIV time-averaged map, b) PTV time-averaged map for the left-side video, c) LSPIV time-averaged map, and d) PTV time-averaged map for the right-side video. Image resolution is decreased from $1024 \times 768$ pixels to $640 \times 480$ pixels for LSPIV processing. White areas in panels b) and d) are due to the absence of detected and tracked particles. Black arrows depict velocity vectors. Vectors in panels a) and c) are magnified by a factor of 2 to improve readability.



**Table 2.** Synoptic table for the input PTV and LSPIV parameters for case studies captured on February, 7th, 2015, at 13:21 and on January 2nd, 2015, at 17:10. For each test, the following parameters are reported: number of analyzed images (n. images), image resolution (res.) and acquisition frequency (freq.), the detection correlation threshold (corr. thresh.), radius, the detection intensity threshold (inten. thresh.), the interrogation area (int. area), the minimum PTV correlation coefficient (min. corr.), the neighbor similarity percentage (sim. neigh.), and the LSPIV grid and interrogation window sizes (grid/wind. size).

| | n. images | res. [pix] | freq. [Hz] | corr. thresh. | radius [pix] | inten. thresh. | int. area [pix] | min. corr. | sim. neigh. [%] |
|---|---|---|---|---|---|---|---|---|---|
| | | | | PTV | | | | | |
| | | | | 02/07 13:21 | | | | | |
| Left | 521 | $1024 \times 768$ | 8.2 | 0.4 | 9 | 100 | 60 | 0.4 | 15 |
| Right | 521 | $1024 \times 768$ | 8.2 | 0.4 | 4 | 100 | 60 | 0.4 | 15 |
| | | | | 01/02 17:10 | | | | | |
| Left | 661 | $1024 \times 768$ | 10.21 | 0.4 | 9 | 100 | 40 | 0.4 | 15 |
| Right | 661 | $1024 \times 768$ | 10.21 | 0.4 | 4 | 100 | 20 | 0.4 | 15 |

| | n. images | res. [pix] | freq. [Hz] | grid/wind. size [pix] |
|---|---|---|---|---|
| | | | LSPIV | |
| | | | 02/07 13:21 | |
| Left | 521 | $640 \times 480$ | 8.2 | 64/32 |
| Right | 521 | $640 \times 480$ | 8.2 | 32/16 |
| | | | 01/02 17:10 | |
| Left | 661 | $640 \times 480$ | 10.21 | 32/16 |
| Right | 331 | $640 \times 480$ | 20.43 | 32/16 |



**Table 3.** Synoptic table of the statistics for time-averaged velocity maps.

|  | $v_{\min}$ | $v_{\max}$ | $\overline{v}$ | $\sigma_{\mathrm{v}}$ | $v_{\mathrm{R}}$ |
|---|---|---|---|---|---|
|  | [m/s] | [m/s] | [m/s] | [m/s] | [m/s] |
| 02/07 13:21 |  |  |  |  | 2.13 |
| LEFT |  |  |  |  |  |
| PTV | 0.16 | 2.09 | 1.33 | 0.19 |  |
| LSPIV | 0.01 | 0.23 | 0.16 | 0.04 |  |
| RIGHT |  |  |  |  |  |
| PTV | 0.06 | 8.49 | 2.23 | 1.13 |  |
| LSPIV | 0 | 4.49 | 2.38 | 1.64 |  |
| 01/02 17:10 |  |  |  |  | 0.69 |
| LEFT |  |  |  |  |  |
| PTV | 0 | 2.30 | 0.35 | 0.40 |  |
| LSPIV | 0 | 0.84 | 0.07 | 0.14 |  |
| RIGHT |  |  |  |  |  |
| PTV | 0 | 4.15 | 0.30 | 0.53 |  |
| LSPIV | 0 | 3.45 | 1.38 | 1.54 |  |