# Peer review of "A novel permanent gauge-cam station for surface flow observations on the Tiber river"

_Geoscientific Instrumentation, Methods and Data Systems, 2015_

## Editor Comment (EC1) · A. Benedetto (Editor) · 11 Apr 2016

The paper presents a very ineteresting instrument and method for flow monitoring. The main point is that the measurement station provides monitoring at large scale and with continuous observation of surface flows. This is based on remote acquisition and calibration of video data. In general this approach and technique could have inetersting applications, considering the promising results that are demonstrated in the paper. The first results are sometimes impressive. Now I think that, after this pilot test, the efforts should be spent in arranging a comprehensive system also including other technologies and sensors.

---

## Editor Comment (EC2) · A. Benedetto (Editor) · 14 Apr 2016

Thank you for your kind and detailed reply

---

## Author Comment (AC1) · 14 Apr 2016

Dear Editor,

We are particularly glad for your comment, where you perfectly highlight the potential of the approach and of the gauge-cam station we have installed in the Tiber River.

What is known in the literature as the LSPIV approach for river surface velocity estimation was introduced some years ago; however, in our opinion, we are still far from having a ready-to-use measurement protocol for many reasons. Indeed, several hardware and software drawbacks, such as optimal hardware configuration, illumination conditions, algorithms for object identification and velocity estimation, measurement uncertainty, and computational time, still need to be tackled.

Only with a permanent station, it will be possible to forge a comprehensive video database that is rich enough to cover all possible testing conditions and to optimize algorithms and procedures.

The permanent station is also a precious testbed to be enriched with supplementary sensors, as you correctly underline, to provide additional information besides surface flow velocity. This integrated approach will hopefully lead us to an enhanced comprehension of the river flow dynamics.

---

## Referee Comment (RC1) · Anonymous Referee #2 · 29 Apr 2016

General comments The paper presents a novel fixed gauge-cam station for flow velocity observations. A study case (one minute video) for the Tiber river in Rome is presented, and by applying optics-based algorithms on video data, use and criticalities of LSPIV and PTV techniques are compared.

The topic is of interest for Geoscientific Instrumenation and has practical relevance for the application in hydrology and environmental engineering. The paper clear and well written. I think the reader could be more assisted in some points (especially the figures) and I propose below a short list of integrations or minor changes

1) Please specify in the abstract the orientation of the area investigated (cross sectional vs longitudinal view)

2) Page line 85. Since both systems are provided by the same company, please specify

when you are dealing with the traditional radar+ultrasonic meter system and when you are describing the LSPIV system

3) Page 4, line 110. How are (or could) the two L25 (high angle)/L76 low angle) sensors data integrated?

4) Page 6 lines 193-195. What was the set up and constraints for this calibration procedure?

5) Page 6-7 lines 196-204. Could you further explain with more details the tracking procedure (for instance, the effects of changing the correlation threshold and the radius. Did you perform a sensitivity analysis on it?)

6) Lage 7. Lines 210-214. So are standard deviations referred both to averaging processes over time (different images) and space (spatial heterogeneity)? Is the variability over time negligible compared to the spatial one?

7) Please show in all figures overall flow direction

8) Specifically, it is not clear if (still unrealistic) gradients of measured velocity with PIV are aligned with the flow or not

9) Could you fill table 1 with similar data obtain through the RVM radar?

---

## Referee Comment (RC2) · Anonymous Referee #1 · 30 Apr 2016

The work presents an interesting test of the performances of a system for measuring surface flow velocity through video recording. The idea is not novel, but the application in a real-world field case-study is very useful in order to understand the potential of the approach for operational purposes. The authors have presented in a previous paper the results of a similar application with LSPIV methodology, but in the present work they compare such approach with a PTV algorithm, which performances appear very promising.

The paper is certainly within the scope of the journal, and interesting for both scientists and practitioners, but it needs a few clarifications and in particular:

1) The description of the events is very confusing: they should be three (see p. 2) but in section 4 (Case study) and in the conclusions only one event is cited) A full description

of the three events (and of their main features, that should illustrate different conditions in the three events: are they high or low flow periods? Which are the atmospheric and lighting conditions? Are there any differences in the amount and quality of the natural floating particles? ) must be added in the case study section. And the results are now presented separately in two different tables that should be merged in one and should report exactly the same information for the 3 events, along with an interpretation of the results that highlights the differences in the three events. It follows that, in the present version, a straightforward interpretation of the results is not possible (also sections 4.1 and 4.2, in addition to the conclusions, seem to refer to one event only...)

2) More information is needed on the radar measurements and in particular it is necessary to know which area is monitored by such sensor (it should be shown on a map, to show the position in reference to the areas covered by the videos)

3) The distinction of the left and right images is not very clear to me... (one is with fish-eye and the other one not? Different lighting conditions too?) The perimeters of the two recorded areas should be shown in the same map with the radar-covered area and more explanations are needed to justify the strong difference in the corresponding results.

4) ll. 4-5 p. 7: please explain why the lighting conditions should affect more negatively the LSPIV than the PTV.

5) ll 8-9 p. 7: this sentence is not clear (what do you mean with 'stationary occurrence of tracers? Artificial seeding?)